# The Effects of Regularization and Data Augmentation are Class Dependent

**Randall Balestriero**
Meta AI Research, FAIR
NYC, USA
rbalestriero@meta.com

**Leon Bottou**
Meta AI Research, FAIR
NYC, USA
leonb@meta.com

**Yann LeCun**
Meta AI Research, FAIR, NYU
NYC, USA
ylecun@meta.com

## Abstract

Regularization is a fundamental technique to improve a model's generalization performances by limiting its complexity. Deep Neural Networks (DNNs), which tend to overfit their training data, heavily rely on regularizers such as Data-Augmentation (DA) or weight-decay with hyper-parameters found from structural risk minimization, i.e., cross-validation. In this study, we demonstrate that the optimal regularization's hyper-parameters found from cross-validation over all classes leads to disastrous model performances on a minority of classes. For example, a resnet50 trained on Imagenet sees its "barn spider" test accuracy falls from $68\%$ to $46\%$ only by introducing random crop DA during training. Even more surprising, such unfair impact of regularization also appears when introducing uninformative regularizers such as weight decay or dropout. Those results demonstrate that our search for ever increasing generalization performance —averaged over all classes and samples— has left us with models and regularizers that silently sacrifice performances on some classes. This scenario can become dangerous when deploying a model on downstream tasks e.g. an Imagenet pre-trained resnet50 deployed on INaturalist sees its performances fall from $70\%$ to $30\%$ on class #8889 when introducing random crop DA during the Imagenet pre-training phase. Those results demonstrate that finding a correct measure of a model's complexity without class-dependent preference remains an open research question.

## 1 Introduction

Machine learning and deep learning aim at learning systems to solve as accurately as possible a given task at hand [LeCun et al., 1998, Bishop and Nasrabadi, 2006, Jordan and Mitchell, 2015]. This process often takes the form of (i) being given a finite dataset, a (differentiable) loss function, and a performance measure, (ii) splitting the dataset into train/valid/test sets to optimizing the system's parameters e.g. from gradient updates of the loss on the train set while cross-validating hyper-parameters using the valid set, and (iii) assessing the system's performance on the test set. *As the training set is finite, and the optimal design of the system is unknown, it is common to employ regularization during the optimization phase to reduce over-fitting* [Tikhonov, 1943, Tihonov, 1963] i.e. to decrease the system's performance gap between train set and test set samples [Simard et al., 1991, Chapelle et al., 2000, Bottou, 2012, Neyshabur et al., 2014]. Central to our study is the fact that hyper-parameter selection is done via cross-validation by maximizing the valid set performance with ad-hoc statistics e.g. the average accuracy over all samples for classes in classification tasks.

Cross-validation commonly involves many different types of regularization along with their "strengths" [Goodfellow et al., 2016, He et al., 2021]. Most variants of regularization take one of two forms: Data-Augmentation (DA) and weight-decay. DA is a data-driven and informed regularization strategy that artificially increase the number of training samples [Shorten and Khoshgoftaar, 2019]. As

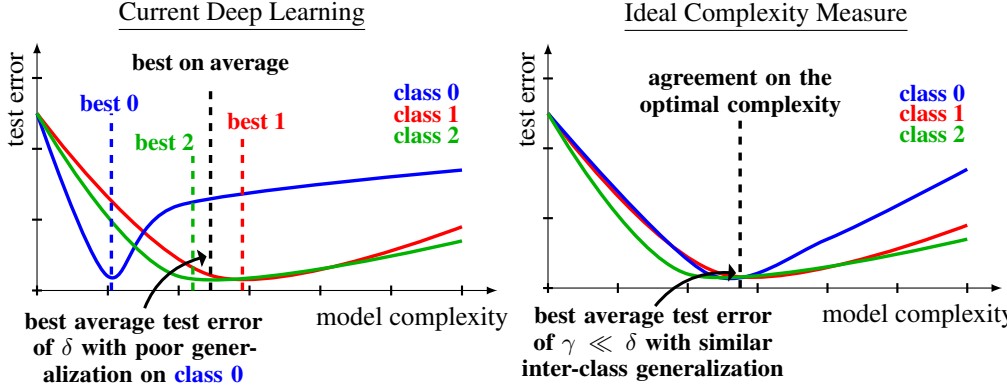

Figure 1: Structural risk minimization minimizes the empirical risk of several models of varying complexity, and selects the one offering the best compromise between under-fitting and over-fitting [Vapnik and Chervonenkis, 1974]. In deep learning, one commonly control the model's complexity by picking different DN architectures and/or by applying different levels and flavors of regularization. The key observation of our study is that *when the model complexity is calibrated by DA (see Figs. 2, 5 and 6), or weight-decay (see Fig. 3), the class-conditional empirical risks do not align between classes i.e. cross-validation produces models that perform well on the majority of classes but arbitrarily poorly on a few of them* as depicted on the left-hand-side. In an ideal setting where the control of the model's complexity is well aligned with the task and model, one would observe the right-hand-side ideal scenario where the same model complexity is optimal for all classes.

opposed to most *explicit* regularizers e.g. Tikhonov regularization [Krogh and Hertz, 1991], also denoted as weight decay, DA's regularization is *implicit* as it is not a function of a model's parameter, but a function of the training samples [Neyshabur et al., 2014, Hernández-García and König, 2018, LeJeune et al., 2019]; although some DA strategies can be turned into explicit regularizers Balestriero et al. [2022]. Nevertheless, a key distinction between DA and weight decay is that DA tends to require more domain knowledge to be successful than weight decay. Most —if not all— of current state-of-the-art employ such regularizers [Huang et al., 2018, Chen et al., 2020b, Liu et al., 2021, Tan and Le, 2021, Liu et al., 2022].

In this paper, we will demonstrate that *when cross-validation is employed to select the regularization settings maximizing the validation performance, a significant bias is introduced into the trained model: the regularized model exhibits strong per-class favoritism i.e. while the average test performance is improved, it is at the cost of producing a model with significant performance drop on some of the classes* as illustrated in the schematic of Fig. 1. For readers familiar with statistical estimation results e.g. the bias-variance trade-off [Kohavi et al., 1996, Von Luxburg and Schölkopf, 2011] or bayesian estimation e.g. Tikhonov regularization [Box and Tiao, 2011, Gruber, 2017], it should not be surprising that regularization produces bias (more details and background provided in Appendix A). In fact, it is beneficial to introduce bias through regularization if it results in a significant reduction of the estimator variance —when one minimizes the average empirical risk. However, the potentially dangerous effect of regularization that this study brings forward is that *the bias introduced by regularization is class-dependent, including on transfer learning tasks*.

To thoroughly validate this observation, we propose a variety of controlled experiments in Section 2. First, we carefully quantify the impact of DA, weight decay and dropout on the per-class performance of a model in Section 2.1, demonstrating that current deep learning finds itself in the scenario depicted on the left of Fig. 1. Then, we consider the task of transfer learning in Section 2.2 where it is again possible to identify again a per-class bias on the target dataset even though the regularization was applied on a different (source) training set. This latter scenario is particularly relevant in current times where it is common to deploy a large pre-trained model on a variety of tasks and raise an important issue: selecting the —on average— best performing pre-trained model can lead to catastrophic individual class performance even on for different downstream tasks.

Our next Section 3 will aim at exploring possible explanations and solutions. First, we will provide a brief theoretical justification on why and when DA can be the cause of model bias (Section 3.1) regardless of the task and data at hand. This will shed light to a first possible issue: the DA parameters that make the transformed input preserve its label information vary depending on the class underlying statistics. In short, DA silently introduces class-imbalance in the training set. We propose a dedicated

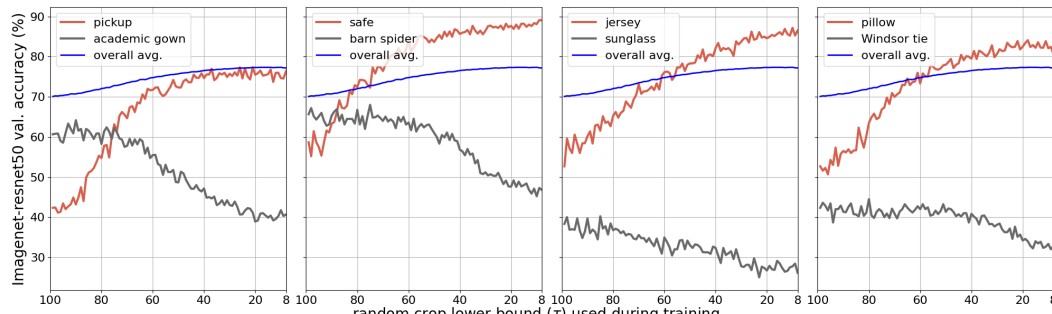

Figure 2: Varying the random crop DA lower bound (x-axis) from 100% to 8% provides greater average test accuracy (blue) but makes the per-class performance fall for some of the classes. Images of each class are provided in Fig. 12, in the appendix. See Fig. 8 for the convnext and ViT experiments. Results obtained by averaging over 20 runs, official PyTorch resnet50 implementation trained on Imagenet with horizontal flip and varying random crop lower bound DA.

analysis of the label-preserving property of DA on different classes and models in Section 3.2. We then take on the task of searching for a possible solution by first reviewing known theoretical studies quantifying the interplay between regularization and bias in Section 3.3. Lastly, we propose some solutions of our own in Section 3.4 built from the gained insights of Section 3.2 using label-distillation and adaptive DA. *All the codebase used to train the various models and to generate the figures is in the supplementary files.*

## 2 Maximizing the Average Model Performance by Cross-Validation Silently Produce Poor Final Performances on a Minority of the Classes

We now turn to the empirical validation of Fig. 1 i.e. quantifying the amount of class-dependent bias caused by DA, weight decay and dropout in various realistic scenarios (Section 2.1). We then demonstrate how the bias introduced by regularization transfers to downstream tasks e.g. when deploying an Imagenet (*source*) trained model on the INaturalist (*target*) dataset in Section 2.2; that scenario is key as it demonstrates the potential harm of selecting the best performing model on the source dataset which could turn out to also be the most biased model against the target dataset class of interest.

### 2.1 Precisely Measuring the Per-Class Effect of Data-Augmentation and Uninformed Regularization with Controlled Experiments

This section aims at quantifying precisely the amount of downward or upward per-class performance shift that came as a result from using DA or uninformed regularization e.g. weight-decay or dropout. In fact, it is crucial to remember that regularization, or any other form of structural risk minimization, improves generalization performances by increasing the bias of the estimator so that the estimator's variance is decreased by a greater amount. However, nothing guarantees the fairness of this bias i.e. for it to be equally distributed amongst the dataset classes. We thus propose a sensitivity analysis by training a large collection of models with varying regularization policies to precisely assess their impact on the class-dependent model bias.

**Data-Augmentation.** DA samples have been known to sometimes disregard the semantic information of the original samples [Krizhevsky et al., 2012]. Nevertheless, DA remains applied universally across tasks and datasets [Shorten and Khoshgoftaar, 2019] as it provides significant performance improvements, even in semi-supervised and unsupervised settings Guo et al. [2018], Xie et al. [2020], Misra and Maaten [2020]. To measure the impact of DA onto per-class performances, we propose in Fig. 2 a sensitivity analysis by training the same architecture on Imagenet with varying DA policies. In particular, we consider a given DA (random crop in this case) and we vary the support of the parameter $\alpha$ which represents how much of the original image is kept in the crop (examples at the top of Fig. 5). We train multiple Deep Neural Networks (DNN)s using $\alpha \in [100, \tau]$ with $\tau$ varying from 100 to 8 and for each case, we report our metrics averaged over 20 trained models. We observe a clear relation between increase in the strength of the DA, increase in the average test accuracy overall classes, and decrease in some per-class test accuracies. For example, on a resnet50 Imagenet setting, the accuracy on the "academic gown" class goes from 62% to 40% steadily as $\tau$ decreases.

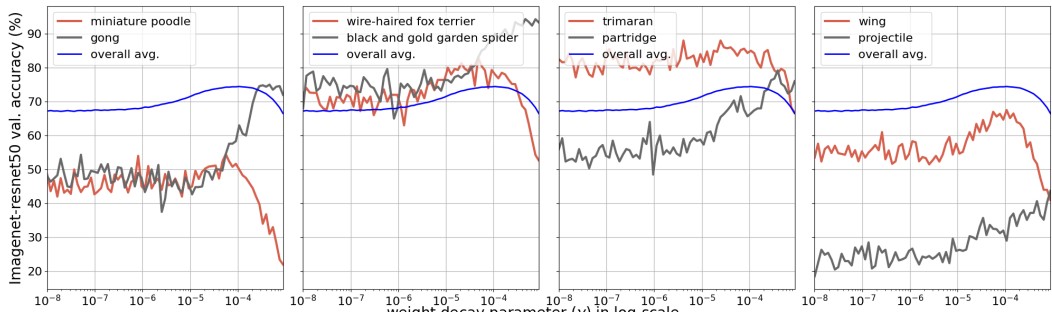

Figure 3: Varying the amount of weight decay, an uninformed regularizer employed throughout, surprisingly exhibits similar class-dependent bias as the DA scenario of Fig. 2. Images for each class are provided in Fig. 13, in the appendix. Results obtained by averaging over 20 runs, official PyTorch resnet50 implementation trained on Imagenet with varying weight decay, see Fig. 16 for DenseNet121 results with the same trend and Fig. 8 for the convnext and ViT experiments.

**Uninformed Regularization.** As per the arguments given in Sections 3.1 and 3.2, it would be natural to assume that what makes DA responsible for creating class-dependent bias in DNNs is our misfortune in defining correct augmentation policies. Hence, uninformed weight decay or dropout should behave differently and more fairly. We demonstrate here that such regularizers are also unfair between classes. We thus propose to train multiple models with varying weight-decay and dropout parameters. In our setting, weight-decay is applied to all the DNN parameters except for the ones of batch-normalization layers, as commonly done [Hastie et al., 2009, Leclerc et al., 2022]. We report in Fig. 3 the per-class performance of a resnet50 trained on Imagenet with varying weight decay coefficient $\gamma$ (as was done for DA in Fig. 2) and we observe that different classes have different test accuracy sensitivities to variations in $\gamma$. Some will see their generalization performance increase, while others will have decreasing generalization performances. We further confirm such findings in Figs. 9 and 10 for dropout where the same per-class trend is observed. In short, even for uninformative regularizers such as weight decay or dropout, a per-class bias is introduced, reducing performances for some of the classes. Although weight-decay is one of the most popular regularizer that is uninformed on the data and task at hand, recent studies have demonstrated that techniques such as model pruning —which can be seen as a post-training model complexity reduction i.e. regularization— also produce increased bias towards under-represented features [Hooker et al., 2019, 2020]. More recently, Balestriero et al. [2022] obtained the close-form explicit regularizer of DA from which it is possible to quantify the sample-dependent aspect of DA's regularization. From our findings, it seems that classes sharing the same type of features are thus impacted different by DAs.

**Formal Statistical Test.** To further convey our claim, we now propose a formal statistical test [Neyman and Pearson, 1933, Fisher, 1955] on the hypothesis that the per-class accuracy is significantly higher when DA is applied for each class (details provided in Appendix A.2). We obtain that there is enough evidence to reject the hypothesis with 95% confidence for 4.5% of all the classes, and with 99% confidence for 2.6% of all the classes. Hence there is sufficient evidence to say that the per-class test accuracies is not increased when introducing DA for 4.5% of the 1000 Imagenet classes. We provide in Table 1 the same statistical test but applied on a variety of settings including different architectures (resnet50, densenet121, ViT-small and ConvNext-Tiny) and across the random crop DA and the weight decay controlled experiments. We should highlight however that this is not necessarily a meaningful measure since for example one regularization might not have any negative impact on the classes, but can provide a beneficial gain that is drastically different between classes. Hence, although the per-class performance does not drop by introduce the regularizer, the inter-class performance gap can be increased by it, which is an equally harmful impact.

The next Section 2.2 proposes to study the scenario of introducing a pre-trained model, on a different downstream task to show that the class-dependent effect of regularization remains present and unfair towards specific downstream classes.

## 2.2 The Class-Dependent Bias Transfers to Other Downstream Tasks

The last experiment we propose is to quantify the amount of class-dependent bias that transfers to other downstream tasks, a common situation in transfer learning and in system deployment to the real world [Pan and Yang, 2009]. We thus want to measure how regularization applied during the

Table 1: Percentage of Imagenet classes for which the test set performance (per-class) is statistically not greater when applying random crop DA (or weight decay) as measured by the statistical test from Section 2. We observe that although this measure is highly conservative since a regularizer (DA or weight decay) might now have a negative impact on a per-class performance but still increase the performance gap between difference classes, already, a nonzero proportion of classes are negatively impact by introduce random crop DA or weight decay.

| confidence | resnet50 random crop | resnet50 weight decay | convnext-tiny random crop | convnext-tiny weight decay | ViT random crop random crop | densenet121 weight decay |
|---|---|---|---|---|---|---|
| 90% | 5.9 | 39.4 | 0.1 | 85.5 | 1.2 | 33.8 |
| 95% | 4.5 | 30.3 | 0.1 | 78.8 | 1.0 | 17.1 |
| 99% | 2.6 | 15.3 | 0.0 | 63.9 | 0.7 | 3.3 |

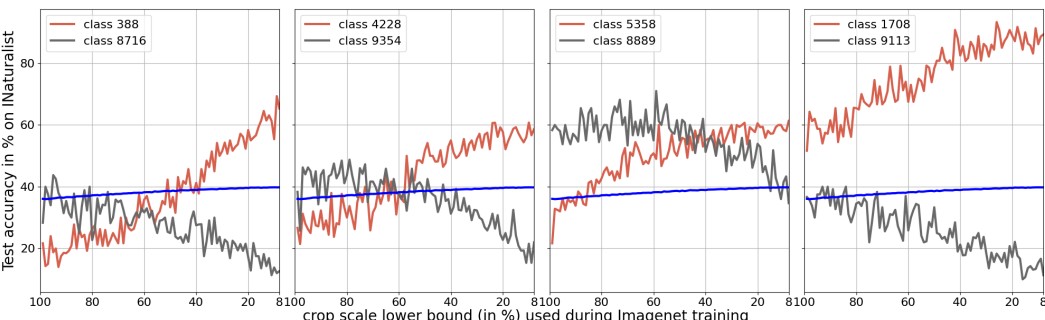

Figure 4: Varying the random crop lower-bound during pre-training (Imagenet) also produces unfair per-class biases on the downstream task (INaturalist). Hence, selecting the pre-trained model with best average test accuracy on the source dataset might result in deploying a model with the worst performance on the classes of interest in the target task. Images for each class are provided in Fig. 17, in the appendix. Results obtained by averaging over 20 runs, official PyTorch resnet50 implementation trained on Imagenet with varying random crop lower bound and transferred to INaturalist with frozen backbone parameters.

pre-training phase on a *source* dataset impacts the per-class accuracy of that model on the *target* dataset.

In order to keep the setting similar to Section 2.1, we adopt a resnet50 model with random crop DA. That model is pre-trained on Imagenet dataset (source) with varying value of $\tau$ (random crop lower bound) and then, the trained model is transferred to the INaturalist dataset [Van Horn et al., 2018] (target) that consists of 10,000 classes. When transferring the model to INaturalist, the parameters are kept frozen, and only a linear classifier is trained on top of it. We report in Fig. 4 the performance of the trained models with varying $\tau$ on different INaturalist classes. We observe once again that the best resnet50 —on average— is not necessarily the one that should be deployed as there exists a strong per-class bias that varies with $\tau$. As a result, picking the best performing model from a source dataset to a target dataset, might leave the pipeline to perform poorly since that model might also be the one that is the most biased against the class of interest in the target dataset.

This result should motivate the design of novel regularizers that do not reduce performances between classes at different regimes. Additionally, due to the cost of training multiple models with varying regularization settings, one might wonder on the possible alternative solutions to detect trends such as shown in Fig. 4 only when given a single pre-trained model.

## 3 Understanding Why and When Regularization Produces Models With Class-Dependent Preferences

The first part of our study (Section 2) empirically validated that DNN regularization produces unfair model complexity control over different classes, resulting in a model performing poorly on a few of the classes although being highly performing on average. We now provide in Sections 3.1 and 3.2 some intuition on why DA can be a source of bias regardless of the task, dataset and model at hand. Then, Section 3.3 reviews existing works trying to confront regularization and model bias, and as we will see, an out-of-the-box solution does not seem to exist when there are only a few classes suffering from regularization (Section 3.4).

## 3.1 A Data-Augmentation Policy That is Not Label-Preserving For All Classes Will Create Class-Imbalance Performances

To provide a simple explanation on how DA causes bias in a trained model, we propose the following derivation that holds for any signal e.g. timeseries, images, videos. Without loss of generality, we will consider here the $\ell_2$ loss which was shown to perform as well as the cross-entropy even on Imagenet [Hui and Belkin, 2020].

**Dataset notations.** Given a sample $x \in \mathcal{X}$ with $\mathcal{X} \subset \mathbb{R}^D$, we consider $y \triangleq f^*(x)$ to be the ground-truth target value. Hence our hope is to learn an approximator $f_\theta$ that is as close as possible to $f^*$ everywhere in $\mathcal{X}$, although we only observe a finite training set $\mathbb{X} \triangleq \{(x_1, y_1), \ldots, (x_N, y_N)\}$. Given an output vector $u$ we also define the level-set of a mapping $f$ to be $\{x \in \mathcal{X} : f(x) = u\}$.
**Data-Augmentation notations.** Additionally, one employs a DA policy $\mathcal{T} : \mathbb{R}^D \times \mathcal{K} \mapsto \mathbb{R}^D$ such that given a transformation parameter $\alpha \in \mathcal{K}$, $\mathcal{T}_\alpha(x)$ produces the transformed view of $x$. Often, one also defines a density $p$ on $\mathcal{K}$ that helps in sampling transformation parameters that are a priori known to be the most useful.

**Theorem 1.** Whenever the transformations produced by $\mathcal{T}_\alpha, \forall \alpha$ do not respect the level-set of $f^*$, and whenever the model has enough capacity to minimize the training loss, the DA will create irreducible bias in $f_\theta$ as in

$$\underbrace{\sum_{(x,y) \in \mathbb{X}} \mathbb{E}_\alpha \left[ \|y - f^*(\mathcal{T}_\alpha(x))\|_2^2 \right] > 0}_{= 0 \text{ iff the DAs of } x \text{ are on the same level-set of } f^*} \text{ and } \underbrace{\sum_{(x,y) \in \mathbb{X}} \mathbb{E}_\alpha \left[ \|y - f_\theta(\mathcal{T}_\alpha(x))\|_2^2 \right] = 0}_{\text{zero training error}} \implies \text{ biased } f_\theta. \quad (1)$$

The main idea of the proof, provided in Appendix B, is to show that if a transformation does not move samples on the level-set of the true function (left-hand-side of Eq. (1)), then $f_\theta$ will learn a different level-set (since it has 0 training error), and thus $\|f^* - f_\theta\| > 0$ i.e. $f_\theta$ is biased.

Whenever the left-hand-side of Eq. (1) is 0, the DA is denoted as *label-preserving* [Cui et al., 2015, Taylor and Nitschke, 2018]. From the above, we see that *unless the target $y$ associated to $\mathcal{T}_\alpha(x)$ is modified accordingly to encode the shift in the target function level-set produced by $\mathcal{T}_\alpha$, any DA that is not label-preserving will introduce a bias.* Some DAs propose to incorporate label transformation i.e. not only $x$ but also $y$ is augmented to better inform on the uncertainty that has been added into $\mathcal{T}_\theta(x)$. This is for example the case for MixUp [Zhang et al., 2017], ManifoldMixUp [Verma et al., 2019], CutMix [Yun et al., 2019] and their extensions.

Our goal in the next Section 3.2 is to demonstrate how DAs such as random crop, color jittering, or CutOut are only label preserving for some values of $\alpha$ that vary with the sample class. As a consequence, while the use of the DA improves the average test performance, it is at the cost of a significant reduction in performance for some of the classes.

## 3.2 The Same Data-Augmentation can be Label-Preserving or Not Between Different Classes

In the previous Section 3.1 we provided a general argument on the sufficient conditions for DA to produce a biased model. We hope in this section to provide a more concrete example that applies to current DNN training. To that end, we will demonstrate that a DA can be label-preserving or not depending on the sample's class, hence, since the same DA policy is employed for all classes, the augmented dataset will exhibit a class-imbalance in favor of the classes for which the DA is most label-preserving.

To measure by how much a given DA, $\mathcal{T}_\alpha$, is label-preserving, we propose to take 6 popular architectures that are pre-trained on Imagenet [Deng et al., 2009] from the official PyTorch [Paszke et al., 2019] repository, and to evaluate their performances for varying DA settings (top of Fig. 5). We observe that considering the dataset as a whole is not a good indicator of the optimal DA value to employ since per-class accuracy performance (bottom of Fig. 5) vary drastically. For example, for some classes, any level of transformation $\alpha$ can produce augmented samples with enough information to be correctly classified, while for other classes, even a small amount of DA makes the samples unpredictable.

To further ensure that the observed relation between label-preservation, sample class, and amount of transformation $\alpha$ is sound, we provide in Fig. 6 the per-class test accuracy on different models, all exhibit the same trends. In short, we identify that *when creating an augmented dataset by applying the*

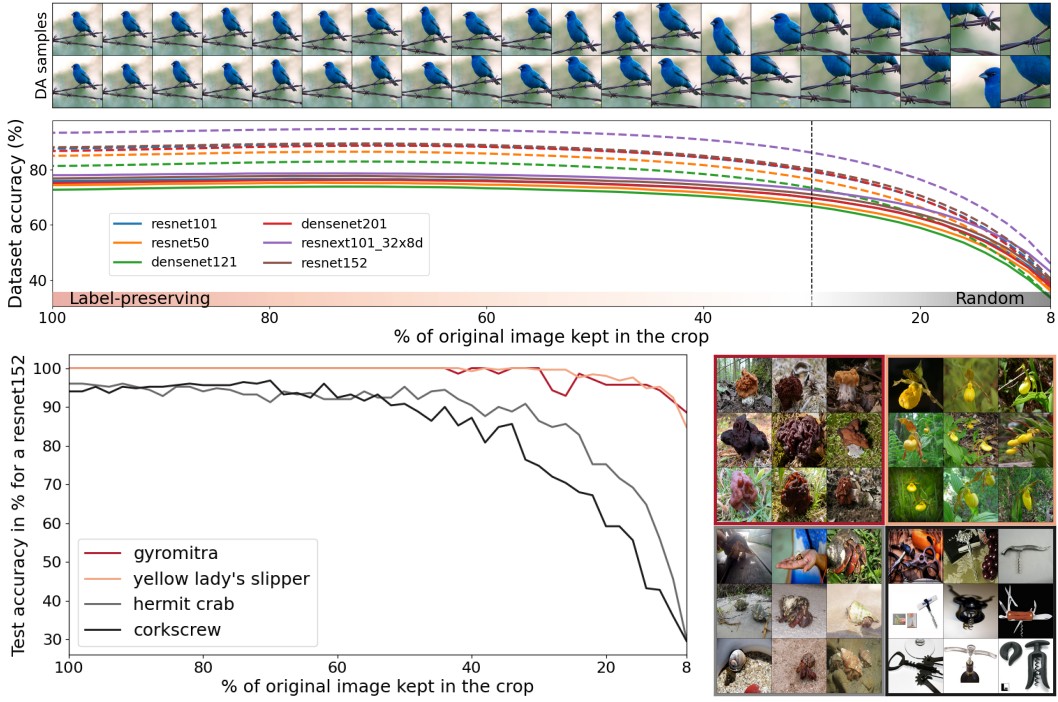

Figure 5: **Top:** examples of an augmented image of class "bird". **Middle:** average accuracy (train set in dashed line and test set in plain) on Imagenet, using 6 popular architectures. **Bottom:** per-class performances from the middle scenario along with 9 images of the corresponding classes. We observe that the random crop DA seems to loose its label-preserving property on average when less than 30% of the image is kept in the crop but looking at the per-class performance, we observe that such DA can be label-preserving with only 8% of the original image for some classes, while for other classes the label information starts to reduce at around 50%. Results obtained from the official Imagenet pre-trained PyTorch models. CutOut and ColorJitter cases are provided in Figs. 14 and 15 and exhibit the same trend.

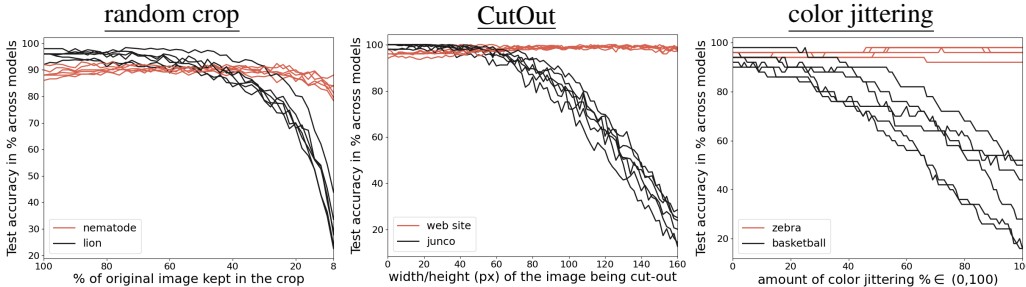

Figure 6: Reprise of the bottom left of Fig. 5 for three different DAs (**each column**) and using the same 6 popular architectures (**different lines**). We observe that across DAs, different architectures agree on the label-preserving regimes for $\mathcal{T}_\alpha$ i.e. even an ensemble of model would not reduce the class-dependent bias of the final prediction. Results obtained from the official Imagenet pre-trained PyTorch models.

*same DA across classes, the number of per-class samples that actually contain enough information about their true labels will become largely imbalance between classes, even if the original dataset was balanced.* Any model trained on the augmented dataset will thus focus on the classes for which the DA is the most label-preserving.

Beyond the above intuitive and natural understand we obtained in term of data-augmentation, there exists theoretical studies that we propose to summarize below

## 3.3 Do Existing Studies Provide Answers Into the Inter-Play Between Regularization and Class-Dependent Bias?

We explored in Section 3.2 a possible explanation of the class-dependent bias we observed in Section 2 hinting at the need to use class-dependent DA. We propose here to summarize existing studies that

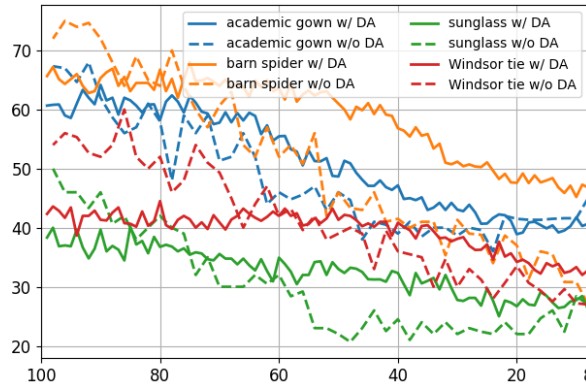

Figure 7: Reprise of Fig. 2 but now implementing the class-dependent DA as prescribed in Section 3.3 i.e. we only apply DA (random crop) to the classes that see their per-class accuracy improve when this DA is employed during training.. We observe here that the bias of the model introduced from random crop on the majority of the classes spills-over to the minority of the classes that do not benefit from that DA, even though such classes never received that DA during training. Results are averaged over 5 runs and employ the official resnet50 implementation trained on Imagenet with horizontal flip but a per-class random crop DA.

are attempting to better understand the impact of DA and regularization in general onto a model's performance and bias. Then, Section 3.4 will explore their application.

**Input independent DA exacerbates the bias already present in a dataset.** Especially relevant to our results is a recent result of Xu et al. [2020]. In this work, it was theorized that when the underlying dataset contains inherent biases, training on the original data is more effective than employing an i.i.d. DA policy, i.e. applying the same random augmentation to all samples/classes, to produce an unbiased model. In short, the DA exacerbates the already present biases and makes the trained model further away from the unbiased optimum. The difficulty of this result lies in defining bias for real images. As per our experiments from Section 3, we observe that bias can take many form e.g. one class might naturally represent its object always under the same angle. This is particularly true say for boats which are rarely captured upside-down. Hence, simply having classes with different natural statistics could be enough for Xu et al. [2020] to prohibit the use of DA in current datasets. Furthermore, Raghunathan et al. [2020] obtained a surprising result combining both DA and regularization. In that case, it was found that the minimum norm interpolant on the original + DA dataset could have a larger standard error than the minimum norm interpolant on the original dataset alone, even when using label-preserving DA. However, this phenomenon only occurs as long as the model remains over-parametrized, even when considering the original+DA dataset.

**Input dependent DA can reduce sample/class bias.** Recall that Section 3.2 brought forward one possible explanation on how DA can be the source of training-set by introduce class-imbalance due to the same DA being label-preserving for some classes and not for others. From this, a direct solution would be to adapt the "strength". This solution, formalized in Xu et al. [2020], consists in measuring the bias of a model and adapt the DA policy accordingly to correct it. This has been done in different flavors e.g. in McLaughlin et al. [2015], Iosifidis and Ntoutsi [2018], Jaipuria et al. [2020]. One limitation of this direction is that it requires to estimate a model bias and adapt the DA accordingly which can be challenging for large scale models.

**Learned DA e.g. from a GAN can produce even more class-dependent bias.** One natural extension of the hands-on adaptivity of a DA to a measured model bias would be to learn a DA to maximize a model's performance, for example. This line of work has led to many learn DA policies e.g. using Generative Adversarial Networks [Hu and Li, 2019]. Yet, it has been shown that learning a DA to maximize some aggregated measure of performance will produce even more bias in a model [Hu and Li, 2019]. In fact, and as per the controlled experiments from Section 2, the learned DA will entirely disregard a minority of the classes if it means that the produced DA can drastically improve performances on all others, effectively maximizing the average performances.

**Inherent tradeoff between DA and model robustness.** In addition to the implication of DA into bias, there exists an intertwined relationship between DA and model robustness. In fact, even assuming the use of perfectly adapted DAs, there exists an *inherent tradeoff* between accuracy and robustness that holds even in the infinite data limit [Tsipras et al., 2018, Fawzi et al., 2018, Zhang et al., 2019]. For example, Min et al. [2021] proved in the robust linear classification regime that (i) more data improves generalization in a weak adversary regime, (ii) more data can improve generalization up to a point where additional data starts to hurt generalization in a medium adversary regime, and that (iii) more data immediately decreases generalization error in a strong adversary regime.

### 3.4 Class-Dependent Data-Augmentation Seems Insufficient for Performance Recovery

We observed in Section 2 that DA could lead to disastrous per-class performances on a minority of classes. We now propose to implement one solution from Section 3.3 that consists in simply not applying the harmful DA to the classes suffering from it. As will become clear, applying the DA on all other classes will be enough to skew the training of the model preventing any performing recovery on those classes.

To motivate the need for a better control of the per-class performance of a model, we first present an illustrative argument. A standard resnet50 on Imagenet reaches $77.14\%$ top-1 with a random crop lower bound of $8\%$. Using precise cross-validation, one could reach $77.29$ by using a random crop lower bound of $10\%$ instead of $8\%$ on all classes. Yet, and most interestingly, if one were able to get the best per-class performance —as per varying the lower-bound as in Fig. 2 and picking for each class the best per-class performance of any of the models— one could reach $79.37$. Beyond pure average test performance, controlling the worst-case per-class performance is of crucial importance for fairness [Du et al., 2020, Veitch et al., 2021]. We thus explore one of the solution that we reviewed in Section 3.3 that consists in only applying the random crop DA to the classes whose test performances increased with the DA's level. We obtain in Fig. 7 that such class-specific strategy is not sufficient to guarantee that the classes negatively impacted by the random crop DA see their performance to be constant across the DA level applied to all the other classes. This finding is also supported by our weight decay experiment in Figs. 3 and 16 in which the regularization of the DN impacted classes differently. In fact, first notice that applying weight decay only when seeing some specific classes would simply amount (on average) to reducing the weight decay hyper-parameter proportionally to how many classes are considered to be without regularization. In a similar fashion, DA produces an implicit regularizer [Balestriero et al., 2022] and applying a class-specific DA level reduces the impact of the implicit regularizer. But since the amount of classes for which we do not apply random crop is quite small compared to the total number of classes (between 1% and 5%), it means that the DA's implicit regularizer remains nearly the same and thus the model's bias is nearly the same regardless if that DA is applied or not onto those classes. This is what we observe, applying the random crop DA to the classes that benefit from it is enough to bias the model and degrade the performances on some classes at the same pace than when applying the DA to all classes unconditionally (compare Figs. 2 and 7).

As a result, we observe that no readily and easily implemented solution provides us with a strategy to prevent deep learning to fall into the scenario depicted on the left of Fig. 1. Those findings however motivate the search of novel model complexity controls that is fair among classes. From a more theoretical viewpoint, it might also be possible to better understand if even such a fair per-class model complexity could exist, which is not clear as natural image classes tend to have inherently different statistics.

## 4 Conclusions and Limitations

We proposed in this study to understand the impact of regularization, in particular data-augmentation and weight decay, into the final performances of a deep network. We obtained that the use of regularization increases the average test performances at the cost of significant performance drops on some specific classes. By focusing on maximizing aggregate performance statistics we have produced learning mechanisms that can be potentially harmful, especially in transfer learning tasks. In fact, we have also observed that varying the amount of regularization employed during pre-training of a specific dataset impacts the per-class performances of that pre-trained model on different downstream tasks e.g. going from Imagenet to INaturalist. Lastly, commonly prescribed solutions e.g. class-dependent data-augmentation do not seem to help indicating that the sole use of an augmentation on some classes is enough to bias the model on all classes. Hence, there remains a vast research area to explore in order to turn deep learning model selection from the current regime to a more ideal one (left to right of Fig. 1).

The main limitation of this work is its focus on computer vision datasets and models. It is possible that our observation will be further conveyed in other regimes or not, and we leave such analysis for future work.

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
