# OpenReview forum: "The Effects of Regularization and Data Augmentation are Class Dependent"
_NeurIPS.cc/2022/Conference — NeurIPS 2022 Accept_

### Official Review · Reviewer_aqbS · 2022-07-11

**Rating:** 7
**Confidence:** 4
**Soundness:** 3 good
**Presentation:** 3 good
**Contribution:** 3 good

**Summary:**

The paper investigates the per-class effect of some common regularization techniques including data augmentation (DA), weight decay, and dropout on neural classifiers.  Experiments across several models and two classification tasks are conducted to show that the popular approach of maximizing mean model performance by cross-validation has heterogeneous effects on different classes. The paper combines intuitive theoretical and empirical evidence to show that the class-dependent label-preserving regime may be the cause of the harmful effects on some classes by inducing class-imbalance for augmented data.

**Questions:**

Questions and suggestions along with the contexts can be found in the “weaknesses” section.

**Limitations:**

The authors discussed the limitations of the work. Please see weakness 1 for some suggested discussion.

**Strengths And Weaknesses:**

Strengths:

1. Overall the paper is well written and easy to follow.

2. The question of interest is well motivated as regularization techniques are a crucial part of the success of modern neural models which are usually overparameterized.

3. The authors provide a good discussion of several lines of related work.

4. There is a good combination of intuition, math, and experimental validation which offers a well-balanced package.

5. Experimental designs are simple yet well controlled to demonstrate the class-dependent effects of regularization in a clear way.

6. Experiments are conducted with a variety of realistic neural models on large and popular datasets/tasks including ImageNet (supervised classification) and iNaturalist (supervised transfer learning).

7. A simple fix with class-dependent DA is evaluated in an attempt to recover the performance.

Weaknesses:

Minor:

1. Despite the fact that pre-trained models of a diversity of architectures are used to observe the label preserving regimes (and they seem to agree qualitatively), I believe that it should be more clearly stated that the observed label-preserving regimes represent those of a CNN family, which may deviate the “true” regime.
Some newer models with patchified input such as ConvNext and Swin Transformers may show different behaviors under the same DA transforms. Additionally, one potentially better empirical estimate may involve repeating the experiments with human observers which are arguably closer to “ideal” observers.

2. The paper can be further strengthened if a practical strategy that recovers the harmful effects of DA on certain classes was proposed.

---

> ### Author Response · Authors · 2022-08-02
> **Answer to Reviewer aqbS**
>
> We thank the reviewer for their insightful remarks and suggestions, and for the positive comments. We believe that our general answer and the associated edits answer the reviewer's concerns.
>
> We have made sure to add experiments on ViT-small and ConvNext-Tiny for the random crop DA (for both) and for the weight decay scenario (for the ConvNext). To be sure to obtain the empirical results by the rebuttal's deadline we have done those experiments with a coarser grid of DA/weight decay parameter but the trend is already easily identifiable. We will be sure to make the formatting of those figures identical to the ones on resnet50/densenet for the final version of the paper.
>
> We have reworked the last section to emphasize that our goal was not to find a solution but rather to demonstrate that the most common of the existing solution (class adaptive DA) fails at solving the issue since the classes negatively impacted by the DA are only a few percentage of the total number of. classes. Hence, the bias imposed onto the architecture by the application of the DA on all other classes remain strong enough to impact those minority classes. We believe that this specific scenario has not yet been thoroughly studied in the literature and yet presents a potential harmful impact for downstream task deployment of pre-trained model.
>
> We hope that those changes answer the reviewer's concerns and we are happy to continue any discussion during the discussion period that is about to start.

---

> > ### Comment · Reviewer_aqbS · 2022-08-08
> > **Response to authors**
> >
> > Thanks for updating the experiments and reworking the last section. As I believe that this was already a strong paper and my concerns were minor, I will keep my original rating.

---

> > > ### Author Response · Authors · 2022-08-09
> > > **Answer to Reviewer aqbS**
> > >
> > > We are delighted that the reviewer found our answers and revision useful to address their comments. We also thank the reviewer for their appreciation of our submission.

---

### Official Review · Reviewer_fQNv · 2022-07-11

**Rating:** 4
**Confidence:** 4
**Soundness:** 2 fair
**Presentation:** 4 excellent
**Contribution:** 2 fair

**Summary:**

The paper set up a hypothesis that implicit regularisation via data  augmentation (in the sense of Poggio's stability) and explicit Tiknonov regularisation (L2 is investigated) penalise different classes by different degrees. This implies that the average test accuracy is not a reliable measure of performance across all class alternatives. It also discovers that data augmentation may even not be label preserving, which it studies in the context of imbalance in the class's support in terms of labels, but it also opens up possibilities of adversarial data augmentation.

**Questions:**

1. If the plot on the RHS in Fig. 1 is contrived to present an ideal case, it puts the rest of the results in question. If it is, mentioning so would help.

2. Both the regularisation strategies investigated induce smoothness. It makes you wonder  if classes are being selectively regularized, what happens with a choice of regulariser that does so by design and produces sparse representations.


**Limitations:**

Although the paper keeps prescriptions out of scope, it may have built upon suggestions made in the work of Raghunath et al. on when and how much data augmentation is suitable.

ALso,  it does not discuss what discuss what proportion of classes are adversely affected by both forms of regularisation, if it is but 1% of classes, the solution may worsen generalisation in the vast majority of classes. Both data augmentation and L2 reguarisation are widely used methods to help with generalisation, and knowng this seems fairly consequential.


**Strengths And Weaknesses:**

The whole set of ideas being investigated esp. the  transferrability to downstream tasks seem to be original although nonetheless a natural extension to the questions posed by Xu and Raghunathan. I would have had more confidence in the hypotheses if 1.the manner in which unfavoured classes were identified was systematised. At the moment, a few hand-picked classes are demonstrated upon. 2. We had quantified support from more datasets and models to rule out bias from them. Otherwise, the paper is delightfully well-explained and well-structured.

---

> ### Author Response · Authors · 2022-08-02
> **Answer to Reviewer fQNv**
>
> We thank the reviewer for their insightful comments and suggestions. We believe that our general answer and the associated edits answer the reviewer's concerns except for a few points that we discuss below.
>
> We have made sure to add to the plot in Fig. 1 that the RHS is an ideal setting that we should aim for.
>
> We have added in the appendix a thorough list of tables depicting the impact on the per-class accuracy of various scenarios to be sure that the figures do not demonstrate any hand-picked cases. We have also added a convnext and ViT experiment to also be sure that the architectures on which this effect appears are not hand-picked. Lastly, we believe that our addition of the systematic statistical test will make our submission more thorough.
>
> We have added the percentage of classes impacted by the DA and regularization. As pointed out by the reviewer, the percentage of classes impacted by DA/regularization is between 1% and 5% and is thus not a major problem when looking at average test performance. However this point is mostly our claim. Because we are focusing on average performance, we disregard a few of the classes' performances, and this scenario can prove itself disastrous when deploying a pre-trained model for other downstream tasks. Especially since we have shown that this per-class bias transfers to other datasets from a pre-trained model.

---

> > ### Comment · Reviewer_fQNv · 2022-08-08
> > **No further questions**
> >
> > Thank you for the explanation to small yet non-insignificant points needing clearing.

---

> > > ### Author Response · Authors · 2022-08-08
> > > **Answer to Reviewer fQNv**
> > >
> > > We are grateful to Reviewer fQNv for not only acknowledging our revision and answer, but also for finding our updates satisfactory. We strongly believe that those suggested changes have made our submission stronger.
> > >
> > > Although the reviewer stated that our updates have clarified important points of our submission, the reviewer maintained their score of 4.
> > > If this is due to any remaining concern or unanswered question, we remain at their disposition to further comment and edit our submission to help improve our submission until the reviewer finds it satisfactory. We believe that the results we have obtained through extensive empirical analysis in our manuscript would be helpful to many in the community and thus would appreciate any pointers from the reviewer to help us improve the (possibly remaining) reviewer's concerns.

---

### Official Review · Reviewer_Yj6j · 2022-07-11

**Rating:** 4
**Confidence:** 4
**Soundness:** 3 good
**Presentation:** 3 good
**Contribution:** 2 fair

**Summary:**

The paper builds around the fact that data augmentation (DA) can harm the accuracies over certain classes despite improving the accuracies overall and offers discussions on several related topics. Some of the discussions are making importance contributions to the community, while other discussions seem fairly straightforward.

**Questions:**

The overall paper is a very interesting read and easy to follow. The main message of the paper, that DA will introduce negative effects to certain classes, despite the overall improvement across all the classes, is quite important and inspiring.



**Limitations:**

There are a couple of generic words regarding the societal impact. The authors do not believe discussions of limitations are relevant, which I don't agree with. For example, the empirical study is limited to a couple choice of transformation functions (and datasets), which could potentially limit whether the conclusions obtained are reproduced in other settings. However, the evaluation is not based on this part. Several questions

- The theorem 1 does not seem to be presented in a rigorous manner, thus very hard to evaluate.

    - there is no definition of "level-set" across the paper. I guess it means decision boundaries or the features related to decision boundaries.
    - in the proof, line 599, assumes p is a constant and omits it, but the remaining proof directly assumes $p=1$ instead of omitting it. (dropping the constant does not allow the equality below line 601 to hold, only assuming it to be 1 does)
    - the proof concludes by saying "last equation is always positive" (line 603), but the entire term is always positive throughout the proof, they are always norms, which are by definition positive.
    - I do not see how other choices of loss, such as cross-entropy, will be carried out trivially following this proof sketch, despite the authors suggest so (line 197)
    - these concerns together, since theorem 1 is the only theoretical result of this paper, put a significant question on the formal results of this work.

- While the paper appears to be dense in contents, I feel like some contents are not significant enough to be presented in the main manuscript, such as details in statistical tests, and augmented samples; some can be presented in a more compact manner, such as Figure 5. To conclude, I feel this paper does not contribute a sufficient amount of results as a paper at this tier.

- Section 3.4 also offers an exciting beginning of the investigation, but the investigation ends very immaturely with a shallow attempt with the most obvious remedy. I would suggest the authors to offer substantially more results following the direction of 3.4.
    - the authors suggested "no readily and easily implemented solution" available (line 311) but it seems not even all the solutions attempted as the previous sections are customized into class-dependent version and tried out yet.

Overall, I think this is an exciting read and interesting start of an important investigation, but the results do not seem mature enough yet.

**Strengths And Weaknesses:**

- strengths
   - the discussion on the effectiveness of class-dependent DA is relevant and potentially will inspire the development of methods for a long way.
   - the main message that DA can harm the performances for certain classes while improving the overall performances is fairly important.
   - the paper is backed up with a heavy set of empirical results with seemingly no issues in reproduction.
- weakness
   - several other messages (other than the main messages) seem very straightforward
   - theorem 1 (the only theorem in the main paper) is not presented rigorously enough for evaluation, which puts up some question for the theoretical rigor of the entire paper.

---

> ### Author Response · Authors · 2022-08-02
> **Answer to Reviewer Yj6j**
>
> We thank the reviewer for their insightful comments and suggestions. We believe that our general answer and the associated edits answer the reviewer's concerns except for a few points that we discuss below.
>
> We have carefully went over Thm. 1 and its proof to be sure that all our derivations are thorough and that each concept has been introduced. We have thus introduced the definition of level set and added descriptions in the proof to explain each step. We also have removed our claim that the same derivations can be carried out to other loss functions to avoid misleading the reader as mentioned by the reviewer. We hope that those changes have made our theoretical result more thorough and self-contained.
>
> We have been sure to compress Fig. 5 and captions from all the figures. We have made sure to provide a more thorough empirical validation including the addition of two state of the art architectures: convnext and ViT. We hope that those changes have improved our presentation and made our submission stronger.
>
> We apologize for the formulation of the last section of our paper. Our attempt was to demonstrate that one of the most common solution (class adaptive DA) fails at solving this issue. However we have reworked our formulation to be sure that the sections reads this way: "One of the most common solution is to adapt the DA for each class. But since the classes suffering from that DA are a small percentage of the whole set of classes (<5%) the impact of DA on the model's bias isn't altered by not applying the DA on such a minority of classes.".
>
> We remain available during the discussion period to address any further comment that the reviewer might have.

---

> > ### Comment · Reviewer_Yj6j · 2022-08-08
> > **Response to the rebuttal**
> >
> > Thank you for your response, the author's choice of listing the high-level concept of changes in the theoretical results instead of discussing the details such as typing all the updates thoroughly might not be the most efficient way of discussions.

---

> > > ### Author Response · Authors · 2022-08-08
> > > **Answer to Reviewer Yj6j (and addition of general answer)**
> > >
> > > We greatly appreciate the reviewer raising their concern regarding our revision presentation. We opted to present only high-level changes here while all the precise updates were highlighted in blue in the revised .pdf. However we entirely agree that this might not be the most efficient exposition and we thank the reviewer for raising that issue.
> > >
> > > In the hope to address that issue, we have summarized with more details (in a new general answer) the edits we have performed (which can still be seen in their context in the .pdf in blue). We sincerely hope that this new form will facilitate all the reviewers' assessment of our edits and we remain happy to discuss any further issues or undressed concerns until the end of the discussion periods.

---

### Official Review · Reviewer_MY5V · 2022-07-13

**Rating:** 7
**Confidence:** 5
**Soundness:** 3 good
**Presentation:** 2 fair
**Contribution:** 3 good

**Summary:**

The paper studies the effects of regularization techniques — including data augmentation (DA), weight decay, and dropout — on per-class accuracies in image classification problems. They demonstrate that when we pick hyper-parameters such as the strength of regularization with regards to the average accuracy across all classes, it does not necessarily result in the fairest model; instead some classes are affected negatively. The experiments tests both in the case of directly training image classifiers as well as finetuning a pretrained classifier on a downstream task. It also explored possible explanations and showed that simple solutions such as selectively applying regularization to certain classes don't mitigate this problem.

**Questions:**

Line 37: What about dropout? It is lightly touched in 2.2 but never mentioned in Introduction

Line 44: requires -> require

Line 78-82: consider using footnote for the promise on codebase. Should delete the second part -- it is not quire appropriate to address to reviewers directly in this style in main text.

Line 147: "all the DN parameters except for the ones of batch-normalization layers" -> I think the author try to say only weights and not biases, but this sentence is confusing as parameters include biases.

Line 206: "respect the level-set of f*" -> unclear what level-set means

Line 218-219: since MixUp etc were brought up as possible fixes for label-preserving, should consider adding them to exps to see if they help at all

Line 235-237: "becomes insufficient to predict the correct label on average" -> should define clearly what that means. What's the definition of "unpredictable"? What's the threshold? The vertical dashed line in Fig 5 is never explained.

**Limitations:**

There's no Limitations section in this paper, which is somewhat reasonable as the authors only exposed the problem (effect of regularization on various classes) but didn't propose a solution. However, I still suggest adding a section about potential limitations if a hypothetical solution is proposed.

**Strengths And Weaknesses:**

In general the paper is well written, neatly structured, easy to follow, with comprehensive and solid experimental setup. The authors studied an important problem of class fairness: when we apply techniques with the overall average performance across all classes as a target, do we incur silent unfair and unfavorable effects on certain classes? The angle on regularization is unique and novel, and all experiments are well constructed to prove the author's point.

However, the paper can be improved by shifting the emphasis and reconstructing the experimental presentation structure.

The experiments focus overly on data augmentation. However I think it is not too hard to give the same weight on the other two: weight decay and dropout, both only lightly experimented or mentioned. Only the first experiment (2.1) is "scientific" enough — with t-test results; the others only show cherry-picked classes. The author should consider at least adding t-tests to all other sections. In addition, I think the author should define a metric of "class fairness" (e.g. the variance or gap between best performing and worst performing classes) and measure/show it across regularization strengths. Most figures in this paper only show cherry-picked classes, and while illuminating, are not scientifically convincing.

The presentation could be tightened, e.g. Fig 1-4 have almost the same setup, and a lot of texts in 2.1 - 2.3 are repetitive; maybe consider merging 2.1 and 2.2, and bringing dropout results in from appendix. Transitions in between sections are a little redundant, and can be made succinct if needed to save space for, say, adding t-test results to other sections and adding dropout.

Specifically, the presentation of results could've been much stronger with the following suggested changes:

 - Consider merging 2.1 and 2.2, moving dropout results from appendix to main sections
 - t-test should be applied to all scenarios, not just DA
 - In addition to showing cherry-picked classes, define metrics for class fairness and show them across regularization strength
 - should also explain how the class pairs are picked for figs 1-4
 - For each figure, try starting with a succinct title, instead of details on how results are obtained. Like Figure 2:  “Effect of DA on per-class accuracies” Fig 3 “Effect of weight decaly on per-class accuracies.”
 - annotating in figures what every line is. The blue lines in fig 2-4 are not explained
 - Overall, the text of paper used too much italic/underline/bold formatting
 - “Section” "Figure" etc. before numbers should be capitalized
 - undefined what "DN" is. If it's "Deep Network", I don't think that's a common abbreviation.

---

> ### Author Response · Authors · 2022-08-02
> **Answer to Reviewer MY5V**
>
> We thank the reviewer for their insightful comments and suggestions. We believe that our general answer and the associated edits answer the reviewer's concerns except for a few points that we discuss below.
>
> We have pushed dropout in the appendix as we believe this technique to be less popular than weight decay and data-augmentation in common architectures nowadays. We have made sure however to reference the dropout experiments in the main text to be sure that the reader does not miss them. With the extra page allowed for the camera ready however, we would be happy to move into the main text one of the figure on dropout along with a figure on convnext or ViT.
>
> Beyond this, we believe that all the formatting and styling suggested by the reviewer has been incorporated into our submission, and we greatly thank the reviewer for those suggestions that have made our submission much more thorough and systematic. We remain available for further discussions if any point is left unanswered.

---

> > ### Comment · Reviewer_MY5V · 2022-08-08
> > **Thanks for your response and revision**
> >
> > Thank you. I've read your response and checked the revised version of your manuscript. I believe the paper has been improved. I've raised my rating to 7 Accept.

---

> > > ### Author Response · Authors · 2022-08-09
> > > **Answer to Reviewer MY5V**
> > >
> > > We are delighted that the reviewer found our answers and revision useful to address their comments. We agree with the reviewer that those updates have strengthen our submission, and we are very thankful to the reviewer for raising their score accordingly.

---

### Author Response · Authors · 2022-08-02
**General answer to all reviewers**

We thank all the reviewers for their thorough reading and evaluation of our submission. We are also delighted that everyone found the paper overall well written and interesting, the findings to be relevant to the community, and the experimental results to be thorough.

All the reviewers mostly shared two main concerns:
a writing style that is sometimes redundant and could be made more poignant
a presentation of empirical results that is not always systematic
We thus propose to answer those in this general answer. The remaining comments specific to each reviewer will be answered in a per-reviewer answer. *Changes to the paper are highlighted in blue.*

**Writing style**
Thanks to the reviewers constructive comments, we have been able to improve our manuscript by implementing the suggested styling changes. In particular, we have
- **figures**: (i) shorten the figure captions, (ii) removed redundancy between captions, (iii) added the missing legend entries, (iv) slightly reformatted figure 5 to save space
- **text redundancy**: (i) merged section 2.1 and 2.2, (ii) shorten the per-section introduction sentences, (iii) moved details on the statistical test in the appendix with corresponding reference in the main text
- **text clarity**: (i) added the level-set definition before Thm.~1, (ii) clarified Thm. 1’s proof, (iii) switched from Deep Network (DN) to Deep Neural Network (DNN) abbreviation, (iv) reduced styling (bold/italic)
- **limitations/future work**: (i) added a ”limitations” paragraph in the conclusion, (ii) restated the last section to be clear that we do not attempt to find a working solution but instead exemplify that the common class-dependent DA solution fails in this regime. In fact, since the considered DA is detrimental only for a minority of classes, applying a class-dependent DA is not enough to counter the model bias as applying the DA on the majority of the classes is enough to bias the model regardless of the DA application on the minority of the classes that suffer from it.

**Systematic empirical evaluation**: As pointed out by the reviewers, a systematic empirical validation is crucial to convey our message. Hence, we have implemented the suggested changes among a few additional ones in order to improve our presentation and to remove any doubt on the fact that none of the results/figures are “cherry-picked”. In particular we have
- **systematic class selection**: (i) added a full table providing the statistical tests for various experiments (as opposed to doing the test only for one setting as originally presented). We also show in this table the percentage of classes impacted, (ii) added the full tables in the appendix providing the exact per-class accuracy for each class and for various experiments ensuring that although we only depict a few classes in the figures, all the data is available in the manuscript, (iii) added in the appendix a the description on how the pairs are selected for the main figures
- **Other architectures**: added a convnext-tiny and ViT small experiment for the random crop and weight decay cases to ensure that the obtained insights are not specific to resnet/densenet convolutional style models


**Code and summary statistics release**: We also take this opportunity to emphasize that in addition to releasing the codebase to reproduce all the experiments from scratch, the set of statistics and final accuracies will be part of that codebase for all models. Hence,  anyone will be able to build upon those experiments without requiring to retrain any model.

We believe that those changes have greatly strengthened our submission. We remain open to further suggestions by the reviewers and will be happy to discuss any additional points during the discussion period.

The authors

---

### Author Response · Authors · 2022-08-08
**Kind reminder to raise any further concerns**

We would like to kindly remind the reviewers to raise any further questions or concerns that we might have missed in our comments and revision before the end of the discussion period to allow us to address them correctly. As pointed by all the reviewers' original reviews, we believe our findings to be not only of great importance but also to be valuable for a broad audience. We believe that the reviewers have raised crucial comments (e.g. adding ViT/convnext models and improving the systematic evaluation on all models) that we have addressed and that greatly improved our submission.

We would be happy to address any additional comment that the reviewers might have, and we also thank Reviewer fQNv for going through our answers/edits and for acknowledging that their concerns were answered.

---

### Author Response · Authors · 2022-08-08
**Detailed updates done in our previous revision**

We thank again all reviewers for their insightful comments. We provide here a more detailed list of updates that we performed in our last revision (all are in *blue* in the submission) to help reviewers assess our updates:

- Addition of ViT/ConvNext experiments and more systematic evaluation:
  - **Table 1** we added a table to our submission which demonstrates for each architecture (6 in total) the percentage of classes for which the use of DA/weight-decay provides a statistically significant drop in performance at different confidence levels
  - **Appendix D and E** we report the table of results for each class showing the difference of performance when using or not DA/weight-decay for a few models
  - **Figure 8 in appendix** depiction of the impact of DA/weight decay on ViT and convnext models
  - **line 132-139** added experimental details on using ViT ConvNext and mentioning that although we report some statistics, there are many other possible statistics to be computed depending on what the application is *We
provide in Table 1 the same statistical test but applied on a variety of settings including different
architectures (resnet50, densenet121, ViT-small and ConvNext-Tiny) [...]*

- Rewriting of the last section to ensure that we only point out to the fact that class DA is insufficient because the amount of classes suffering from DA is small (<5%) and thus the bias introduced via DA's implicit regularization is still learned by the model as it is applied on all the remaining (>95%) of the classes
  - **line 275, 276**: *As will become clear, applying the DA on
all other classes will be enough to skew the training of the model preventing any performing recovery
on those classes.*
  - **line 291 to 297**: *In fact, first notice that applying weight decay only when seeing some
specific classes would simply amount (on average) to reducing the weight decay hyper-parameter
proportionally [...] and applying a class-specific DA level
reduces the impact of the implicit regularizer. But since the amount of classes for which we do not
apply random crop is quite small compared to the total number of classes (between 1% and 5%),
it means that the DA’s implicit regularizer remains nearly the same and thus the model’s bias is
nearly the same regardless if that DA is applied or not onto those classes .*

- Implementation of a less verbose style for the captions of the figures and sections' first paragraphs
  - **Figure 2 through 7**: captions have been compressed by not repeating the verbose sentence for the experimental details, and by adding a clear statement on what are the conclusions in each of then
  - **Intro paragraph of Section 2.1**, we reformulated the second sentence and on to *uninformed regularization e.g. weight-decay or dropout [...] However, nothing guarantees the fairness of this bias i.e.
for it to be equally distributed amongst the dataset classes. We thus propose a sensitivity analysis
by training a large collection of models with varying regularization policies to precisely assess their
impact on the class-dependent model bias.*
  - **Intro paragraph of section 3** we added a clearer description on our exploration and why per-class DA is insufficient *An out-of-the-box solution does not seem to exist when there are only a few classes suffering
from regularization (Section 3.4).*
  - **Intro paragraph of section 3.1** *Without loss of generality, we will consider here the $\ell_2$ loss which was shown to perform as well as the cross-entropy even on Imagenet [].*

- Theorem 1:
  - **line 181** introduction of level-set definition
  - **main + appendix** we removed any mention that this result holds for other losses (main and appendix)
  - **in the proof** we have clarifier that although all terms are always non-negative, the approximation error can never be 0 unless the correct DA is specified i.e. regardless of the approximation power (even in the limit) there will always be a non-zero approximation error between f and f*
  - **in the proof** we clarified that we used uniform density on the manifold for the data and discussed how to obtain the same for arbitrary densities
- other minor updates:
  - **Figure 1's caption**: addition that "the RHS is the ideal scenario"
  - **in section 2.1 and line 105** we added *(examples at the top of Fig. 5)* to refer to images with varying random crop lower bound
  - **line 113** we reformulated the comment on which parameters are considered for weight decay by *We thus propose to train multiple models with varying weight-decay and dropout
parameters. In our setting, weight-decay is applied to all the DNN parameters except for the ones of
batch-normalization layers, as commonly done [].*
  - **line 129** details on the statistical tests have been pushed to Appendix A.2 and instead * (details provided in Appendix A.2). * has been put in the main text
  - all abbreviations have been correctly introduced and switched to their standard versions e.g. DN - DNN

---

### Meta-Review · Area_Chair_kJpN · 2022-08-26

**Recommendation:** Accept
**Confidence:** Certain

**Metareview:**

This paper studies a natural and important question: could techniques that improve the average performance hurt performance on individual classes? The paper provides comprehensive experiments to answer this question in the positive, for data augmentation, weight decay, and dropout. The authors have made significant attempts to improve the paper during the rebuttal phase, including more thorough experimental evaluations and added clarity of Theorem 1. This is a good paper and I recommend acceptance.

One minor comment: there's some inconsistency in terminologies. The title suggests that DA is not a regularization, but in the paper DA is referred to as a type of regularization. Please make it consistent throughout the paper.

**Award:**

No

---

### Decision · Program_Chairs · 2022-09-14

Accept